# Auto-Thinking Evocation in Video Reasoning via Multi-Stage Granular Reinforcement Learning: Stable, Controllable

## Abstract

R1-style reinforcement learning (RL) for stimulating stepwise reasoning significantly boosts Video-MLLMs' performance on complex tasks, yet drastically impairs response efficiency for simple ones. To further incentivize the auto-thinking capability, existing methods typically incorporate reasoning mode selection into RL reward designs to implicitly regulate thinking preferences across different tasks. However, these methods demand strict tuning of sensitive hyperparameters and careful data management, frequently leading to single-mode dominance when processing video data. To achieve stable and controllable auto-thinking evocation in video reasoning, we design a multi-stage granular RL paradigm. Specifically, the responding process under auto-thinking can be decomposed into two subtasks: 1) Determining the reasoning mode, and 2) Generating correct answers. Due to the self-regressive property of LLMs, the initial token governs the overall response mode, while subsequent tokens critically influence answer correctness. From this insight, we respectively improve the model's ability on the above two subtasks by conducting decoupled RL training on tokens at different positions with two RL phases, Meta-Cognition Training and Cognition-Aware Refinement. In Meta-Cognition Training, we construct a reasoning strategy dataset to explicitly incentivize suitable starting tokens on different questions, which stably prevents single-mode collapse and achieves controllable thinking preferences. For Cognition Aware Refinement, the learning is fully conditioned on reasoning or non-reasoning modes, specifically improving answer accuracy under both modes. Through multi-stage granular RL training, we significantly enhance the reasoning accuracy while steadily endowing the model with auto-thinking ability. Extensive experiments across multiple video reasoning and perception benchmarks demonstrate that our approach achieves distinct thinking rates while significantly reducing responding overhead, ultimately improving overall performance and establishing new state-of-the-art results with superior performance-efficiency trade-offs.

## 1 Introduction

Inspired by the success of DeepSeek-R1 Guo et al. (2025), reinforcement learning (RL) algorithms have enabled multi-modal large language models (MLLMs) to acquire stepwise reasoning capabilities. Specifically, the reasoning process encapsulated within the <think> </think> blocks significantly improves the responding accuracy for complex problems. However, in practical application scenarios, multi-modal tasks (particularly video understanding) often involve a large number of simple perception problems that do not require reasoning (e.g., "What is happening in the video?"). This leads to a significant "over-reasoning" phenomenon, making it difficult for reasoning models to achieve a proper balance between performance and efficiency in real-world applications. To alleviate the aforementioned issue, research interest has been widely drawn to endowing models with auto-thinking capabilities that enable adaptive selection of appropriate response modes (*reasoning* or *non-reasoning*) based on different problems.

Existing explorations on auto-thinking have achieved certain effects but all come with their own limitations. Specifically, some LLM-based methods Zhang et al. (2025); Fang et al. (2025) incorporate the mode-selection into the overall objective of reinforcement learning (RL) via complex reward

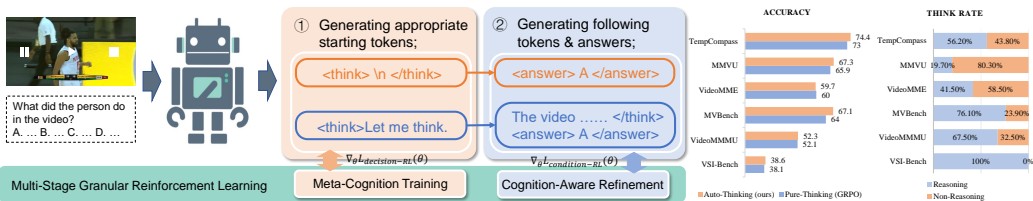

Figure 1: By functionally decoupling tokens at distinct positions, we decompose the model's auto-thinking process into two independent subtasks. We leverage two separate datasets and objectives to optimize the model's performance on these two subtasks through distinct training phases, respectively. After our multi-stage granular reinforcement learning, the model displays drastically divergent thinking rates across different datasets with superior accuracy, demonstrating that our framework stably elicits the model's automatic reasoning capabilities.

designs. However, these methods are confined to the text modality, and they strongly demand sensitive reward value tuning and fine-grained training data control. When trained on multi-modal data, they exhibit high instability and are extremely prone to driving the model into mode collapse. For MLLM tasks, R-4B Jiang et al. (2025) adopts "Bi-mode RL" to avoid the mode collapse issue, yet it lacks supervision over problem-specific reasoning strategy preferences. Additionally, some other approaches Zhan et al. (2025); Team et al. (2025b) introduce an extra analysis process to determine the subsequent response mode. Although this design fosters effective auto-thinking capabilities, the additional overhead required for the analyzing further hinders overall efficiency.

Driving from the above limitations, in this paper we present a novel *multi-stage granular reinforcement learning* method, which *effectively realizes auto-thinking evocation* in video reasoning, while *simultaneously achieving advantages that span training stability, preference controllability, and computational efficiency.*

First, we formalize the model behaviors under **reasoning** and **non-reasoning** modes in Figure 1. Under the response formats that we defined, tokens at different positions exhibit distinct functional roles. Specifically, the starting token has negligible impact on the accuracy of the final answer but directly determines the model's reasoning strategy, while the following tokens (e.g., in reasoning mode) play a decisive role in ensuring the accuracy of the final answer. Thus, the model's responding process can be interpreted as outputting appropriate tokens at different positions to sequentially accomplish two independent subtasks: 1) Generating suitable starting token based on the input question; 2) Generating following tokens and the final answer, guided by the starting token.

Through the above decomposition of the responding process and functional decoupling of tokens, we derive two independent optimization objectives. Building on this, we split the conventional reinforcement learning (RL) process into two sequential stages: Cognition Aware Refinement and Meta-Cognition Training. Two distinct sets of training data and optimization objectives are employed to train tokens at different positions, thereby enhancing the model's accuracy in the two aforementioned subtasks respectively.

For Cognition-Aware Refinement, we utilize vanilla GRPO algorithm Guo et al. (2025) to separately refine the model's responding accuracy under reasoning and non-reasoning modes, where the training is fully conditioned on specific reasoning strategies. To achieve this, we embed the triggering initial token into the prompt sequence to force the model to consistently generate subsequent outputs in specific modes for any input question. Among the GRPO training stage, a critical distinction remains in that the starting token of the response is excluded from gradient computation. This design ensures that the Cognition-Aware Refinement promotes the mode-specific capacity without disrupting the model's inherit automatic thinking strategies.

For Meta-Cognition Training, we explicitly define reasoning preferences for different questions and manually construct training data to supervise the model in generating appropriate query-specific starting tokens. During this stage, the gradients are focused on the starting tokens. Concurrently, we introduce an auxiliary objective to ensure the model successfully transitions into the reasoning/non-reasoning mode based on different starting tokens while adhering to the overall format constraints.

After this stage, the model's selection of reasoning strategy shifts from stochastic to query-aware, exhibiting notably different Chain-of-Thought (CoT) activation rates across diverse datasets.

Compared to existing approaches, our proposed multi-stage granular RL framework eventually leads the model to higher efficiency and performance. In the crucial challenge of auto-thinking evocation, our method exhibits several distinct advantages: First, the model's autonomous reasoning preference for specific question types is **controllable**. By explicitly constructing strategy training data, we can directionally adjust the reasoning behavior based on task requirements. Second, the Meta-Cognition Training process is highly **stable**. The ratio of supervision signals for reasoning and non-reasoning modes can be manually tuned, which greatly mitigates the risk of the model converging to a single fixed reasoning pattern (a common limitation in prior unified RL frameworks). Third, the method is computationally **efficient**. It achieves high-performance autonomous reasoning without requiring additional auxiliary analysis processes, significantly reducing the overall response overhead. Furthermore, the response modes are **configurable**. By embedding different starting tokens into the prompt, we can manually switch the model's response mode among three options: *auto, reasoning, and non-reasoning*. Notably, the functional decoupling of response tokens with distinct optimization objectives represents a highly insightful design. This paradigm of shifting from single-objective to multi-objective optimization in training substantially enhances the controllability of model behavior during post-RL fine-tuning. It effectively achieves auto-thinking evocation and can also transfer to other complex tasks (e.g., multi-step visual reasoning Su et al. (2025); Zheng et al. (2025)).

We evaluated our method on multiple video understanding/reasoning datasets. The experimental results demonstrate that our approach endows the model with robust autonomous reasoning capabilities, showing significant distinct thinking rates across different benchmarks, which drastically reduces the overall token overhead while further improves performance, achieving a more favorable performance-efficiency trade-off compared to other methods.

# 2 RELATED WORKS

**(Multimodal) Large Language Model Reasoning.** The advent of OpenAI's O1 model Jaech et al. (2024) has ignited a surge of research endeavors within the community, with a particular emphasis on augmenting the complex reasoning capabilities of large language models (LLMs) Wei et al. (2022); Yuan et al.; Zhang et al. (2023). Early methodologies predominantly employed dense supervision over the reasoning process Gao et al. (2024); Li & Li (2024), aiming to directly enhance the reasoning prowess of these models. Differently, the remarkable success of DeepSeek-R1 Guo et al. (2025) has paradigmatically demonstrated the efficacy of rule-based reinforcement learning (RL) in bolstering reasoning capability of LLMs, where models autonomously refine their Chain-of-Thought (CoT) processes in the absence of explicit supervision, and manifest potent reasoning behaviors during the RL training phase. This novel paradigm has inspired a wave of subsequent research Zhang et al.; Luo et al. (2025); Liu et al. (2025); Hu et al. (2025a); Team et al. (2025a), with numerous studies Zhou et al. (2025); Yang et al. (2025c); Meng et al. (2025); Huang et al. (2025); Feng et al. (2025) successfully replicated the success of DeepSeek-R1 on Multimodal Large Language Models(MLLMs). While existing methods have effectively endowed MLLMs with robust reasoning capabilities, a critical limitation persists in real-world application that a large proportion of inputs are simple questions that do not require reasoning. Against this backdrop, how to effectively achieve efficient reasoning and attain a more desirable performance-efficiency trade-off has attracted widespread research interest.

**Efficient Reasoning.** Early attempts at efficient reasoning primarily focused on prompt-based reasoning mode selection Yang et al. (2025a); Kang et al. (2025); Aytes et al. (2025); Xu et al. (2025) or adding length penalty to constrain the CoT sequence length Arora & Zanette (2025); Aggarwal & Welleck (2025); Shen et al. (2025); Xiao et al. (2025). To further improve response efficiency, many studies Fang et al. (2025); Jiang et al. (2025); Wang et al. (2025); Zhang et al. (2025); Lou et al. (2025) have attempted to leverage reinforcement learning to endow reasoning models with auto-thinking capabilities that can adaptively adjust reasoning strategies based on input questions. The core of these RL-based approaches lies in designing reward schemes that are influenced by reasoning strategies, aiming to implicitly guide the model to learn appropriate response modes according to answer accuracy. However, such methods often require a sensitive ratio of easy/hard training data and complex reward engineering. When transferred to other models or tasks, they are highly

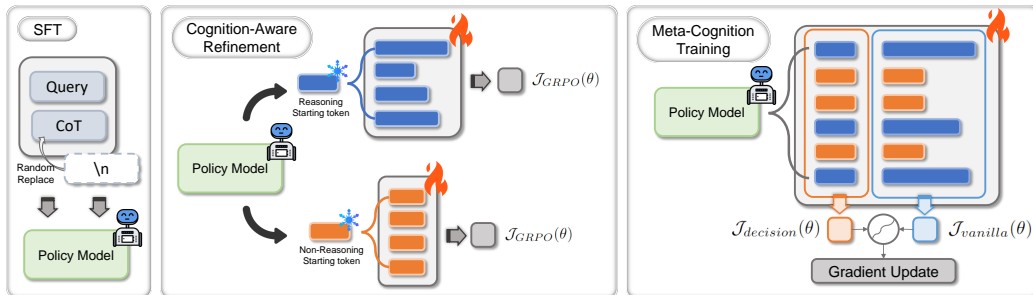

Figure 2: The workflow of multi-stage granular RL to achieve auto-thinking evocation in video reasoning. The SFT stage incorporates thinking-dropout to initially endow the model with both reasoning and non-reasoning abilities. After that, the Cognition-Aware Refinement separately enhance the response accuracy under both modes by GRPO conditioned on specific starting token. The final Meta-Cognition Training stage directly incentivizes the model to produce suitable staring tokens according to the input query, while keeping the subsequent tokens adhere the specific formats.

prone to collapse to fixed reasoning patterns. Some other approaches Zhan et al. (2025); Team et al. (2025b) introduce an additional complexity analysis stage to determine the optimal reasoning strategy. While these methods achieve robust adaptive capabilities, they further increase token overhead, ultimately failing to meet the goal of improving efficiency. In contrast to existing works, we propose a multi-stage, multi-objective RL scheme by decoupling the functional roles of tokens and decomposing the response generation process. This design enables effective auto-thinking evocation while simultaneously achieving the advantages of stability, controllability, and efficiency, addressing the key limitations of prior approaches.

## 3 METHOD

In this section, we elaborate on the detailed workflow of our multi-stage granular reinforcement training. As shown in Figure 2, the entire training process consists of three stages, each with a dedicated objective to optimize the model's responding behavior: 1) SFT with think dropout: to enable the model to initially learn the outputing format under different reasoning strategies without explicit mode labeling. 2) Cognition-Aware Refinement: separately enhancing the model's responding accuracy under specific modes. 3) Meta-Cognition Training: The final stage that endows the model with the ability to adaptively select reasoning strategies based on input questions.

### 3.1 SUPERVISED FINE-TUNING WITH THINKING DROPOUT

First we formalize the model behaviors under **reasoning** and **non-reasoning** modes. Aligning with the convention framework in R1 Guo et al. (2025), the reasoning mode mandates the model to perform stepwise deliberation before answer generation, with its format standardized as `<think>`*the specific thinking process*`</think><answer>`*the result*`</answer>`. In non-reasoning mode, we constrain the model to produce empty thinking block and directly generate the answer, like `<think>\n</think><answer>`*the result*`</answer>`. To initially equip the model with these two distinct response modes, we conduct thinking dropout Wang et al. (2025) during the SFT phase before RL training. Concretely, we randomly replace the annotated thinking process in the SFT dataset with '\n' at a fixed probability (0.5), forcing the model to learn both strategies without explicit mode labeling. After the SFT stage, the model exhibits stochastic output behavior, naturally generating either reasoning or non-reasoning responses.

### 3.2 COGNITION-AWARE REFINEMENT

In the Cognition-Aware Refinement, we adopt the GRPO (Generalized Relative Policy Optimization) algorithm Guo et al. (2025) to separately enhance two key capabilities of the model: 1) the reasoning capability under the reasoning mode, and 2) the ability to directly generate correct answers under the non-reasoning mode. To enable the configuration of model's specific respond-

ing strategy, we ensure that all Chain-of-Thought (CoT) annotations in the SFT dataset start with the trigger phrase "Let me think" during data preparation. After the SFT stage, we can manually control the reasoning strategy for any input question by embedding one of two distinct starting tokens into the prompt: either `<think>Let me think` (to enforce reasoning mode) or `<think>\n</think>` (to enforce non-reasoning mode).

We utilize the Video-Reasoning RL dataset constructed in Video-R1 Feng et al. (2025) for Cognition-Aware Refinement. Notably, to accelerate training efficiency, we perform prior data filtering: we use the post-SFT model to randomly generate 8 responses for each data sample to exclude the overly simple or difficult questions (identified via extremely high or low average rewards), and we finally select 8K multiple-choice questions covering diverse task types as the core training data. To further enhance the model's spatial understanding capability, which is lacking in the Video-R1 dataset, we additionally collect 1K supplementary questions from ScanNet Dai et al. (2017). Ultimately, we construct a dataset $\mathcal{X}_{CT} = \{x_i\}_{i=0}^{9K}$ specifically for Cognition-Aware Refinement.

We conduct two independent GRPO training process to separately improve the model's answer accuracy under different modes, where different starting tokens are embedded in the prompt for each run. In Cognition-Aware Refinement, let $\{o_i\}_{i=1}^{G}$ denote a mini-batch of rollouts sampled from the current policy $\pi_{\theta_{old}}$, the overall objective is defined as:

$$\mathcal{J}_{GRPO}(\theta) = \mathbb{E}_{x,o_i}\Big[\frac{1}{G}\sum_{i=1}^{G}\Big(\frac{1}{|o_i|-l}\sum_{t=l}^{|o_i|}\mathcal{L}_{i,t}(\theta) - \beta\mathbb{D}_{KL}[\pi_\theta(\cdot|x)||\pi_{ref}(\cdot|x)]\Big)\Big] \tag{1}$$

where $\mathcal{L}_{i,t}(\theta)$ denotes the token-level surrogate loss formally given by:

$$\mathcal{L}_{i,t}(\theta) = min\Big(\frac{\pi_\theta(o_{i,t}|x,o_{i,<t})}{\pi_{\theta_{old}}(o_{i,t}|x,o_{i,<t})}\hat{A}_{i,t}, clip\Big(\frac{\pi_\theta(o_{i,t}|x,o_{i,<t})}{\pi_{\theta_{old}}(o_{i,t}|x,o_{i,<t})}, 1-\epsilon, 1+\epsilon\Big)\hat{A}_{i,t}\Big) \tag{2}$$

$\hat{A}_{i,t} = \frac{r-mean(\mathbf{r})}{std(\mathbf{r})}$ represents the group relative advantage for each token, where the reward $r$ is the sum of format reward and accuracy reward. And $l$ is the length of specific starting tokens.

The core distinction between Cognition-Aware Refinement and vanilla GRPO lies in a critical modification: we disable gradient computation for the first $l$ starting tokens in the response sequence. This deliberate design enables the training process to effectively improve the model's mode-specific response accuracy while minimizing unintended perturbations to the model's intrinsic reasoning mode preferences that were initially established during the preceding SFT stage with think dropout. This preservation of intrinsic mode preferences is pivotal for maintaining the stability of subsequent Meta-Cognition Training, where the model will further learn to adaptively select modes based on input questions.

### 3.3 META-COGNITION TRAINING

Existing methods Zhang et al. (2025); Fang et al. (2025) treat the entire response as a monolithic sequence and design answer accuracy-oriented rewards to implicitly learn proper response modes. When reproducing these methods on video-reasoning tasks, we observe two key limitations: (1) such supervision is extremely sensitive to the difficulty ratio of training data, causing the model to easily converge to a fixed reasoning pattern; (2) even if bi-mode responses are achieved via elaborate data difficulty tuning, the model exhibits a uniform thinking activation rate across different test datasets, indicating probabilistic mode selection rather than adaptive choice based on question characteristics. To enable controllable and stable auto-thinking evocation, we explicitly define a reasoning preference criterion and construct a strategy training dataset to directly incentivize the model to generate appropriate starting tokens for specific types of questions.

A question is labeled as thinking "required" or "no-required" based on the probability that the model generates the correct answer with no thinking. Specifically, we force non-reasoning mode on post-SFT Model $\pi_{ref}$ to generate responses for Video-R1 RL data Feng et al. (2025), then compute the probability of generating ground-truth answer tokens upon the model's output logits as $P^{gt} = \frac{1}{|o^{gt}|}\sum_{t\in idx(gt)}\pi_{ref}(o_t|x,o_{<t})$. Questions with $P^{gt} < 0.5$ are labeled as "thinking required" and $P^{gt} > 0.9$ are labeled as "thinking no required". We finally construct a dataset for Meta-Cognition Training $\mathcal{X}_{DT} = \{x_i\}_{i=1}^{4K}$ in a 1:1 ratio.

In the Meta-Cognition Training stage, we randomly generate $\{o_i\}_{i=1}^G$ outputs for each question. Unlike existing methods Jiang et al. (2025); Zhang et al. (2025), we impose no constraints on the sampling process. Based on the model's intrinsic probability distribution, the G samples may either all belong to one same response mode or include both. Using the Meta-Cognition Training data, we directly assign advantages to the starting tokens in responses following the rules below:

$$A_i = \begin{cases} +1 \cdot \gamma, & \text{starting token match mode annotation} \\ -1 \cdot \gamma, & \text{starting token doesn't match mode annotation} \\ -2 \cdot \gamma, & \text{starting token with format error} \end{cases} \quad (3)$$

Notably, the above advantages are directly assigned without group normalization, which avoids the advantage vanish issue in vanilla GRPO that arises when group achieves same rewards. Upon the sampeing results, the meta-cognition objective is defined as:

$$\mathcal{J}_{decision}(\theta) = \mathbb{E}_{x, o_i}\left[\frac{1}{G}\sum_{i=1}^{G}\left(\frac{1}{l}\sum_{t=1}^{l}\mathcal{L}_{i,t}(\theta) - \beta\mathbb{D}_{KL}[\pi_\theta(o_{i,<l}|x)||\pi_{ref}(o_{i,<l}|x)])\right)\right] \quad (4)$$

The meta-cognition objective is focused on $l$ starting tokens, while advantage is in Eq. 3 and the token-level loss $\mathcal{L}_{i,t}(\theta)$ is formatted in Eq. 2.

The meta-cognition objective $\mathcal{J}_{decision}(\theta)$ incentivize the policy model to generate suitable starting tokens according to the mode annotation data, thereby achieves auto-thinking activation. However, our experiments show that computing only decision loss on $\mathcal{X}_{DT}$ leads to gradual chaos in the model's subsequent outputs. To address this issue, we introduce an auxiliary objective target in the Meta-Cognition Training stage to ensure the model generates correct, format-compliant subsequent outputs based on specific starting token. Specifically, we calculate format reward and accuracy reward for subsequent tokens, and compute group average advantage to supervise their generation. Unlike Eq. 1, to further balance the gradient weights of tokens in long thinking responses and extremely short non-thinking responses, we refer to DAPO Yu et al. (2025) and compute token-level averages instead of sequence-level. The detailed calculation is shown in Eq. 5.

$$\mathcal{J}_{vanilla}(\theta) = \mathbb{E}_{x, o_i}\left[\frac{1}{\sum_{i=1}^{G}|o_i| - l}\sum_{i=1}^{G}\sum_{t=l}^{|o_i|}\mathcal{L}_{i,t}(\theta) - \beta\mathbb{D}_{KL}[\pi_\theta(\cdot|x)||\pi_{ref}(\cdot|x)]\right] \quad (5)$$

While $\mathcal{J}_{vanilla}(\theta)$ prevents the model's outputs from collapsing into chaos, numerical discrepancies between the multiple objectives destabilize their joint optimization, manifested as hyperparameter sensitivity, where varying $\gamma$ in Eq. 3 lead to significantly different training results. We attribute this issue to inherent gradient conflicts caused by simultaneous multi-objective optimization. To enhance the stability of Meta-Cognition Training, we design a gated update scheme. Specifically, we decouple the optimization of $\mathcal{J}_{decision}(\theta)$ and $\mathcal{J}_{vanilla}(\theta)$: although both losses are computed in each iteration, only one objective undergoes gradient update. We introduce a hyperparameter $p$ to adjust the overall supervision strength for the two objectives in Meta-Cognition Training. During each parameter update, gradients are computed based on $\mathcal{J}_{decision}(\theta)$ with probability $p$, and $\mathcal{J}_{vanilla}(\theta)$ with probability $1 - p$.

By granular calculation of distinct optimization objectives for tokens at different positions and gated multi-objective optimization, Meta-Cognition Training achieves controllable preference and stable training for auto-thinking activation, enabling the model to reach a more favorable performance-efficiency trade-off.

## 4 EXPERIMENTS

### 4.1 EXPERIMENTAL SETUPS

**MLLM and Datasets.** We adopt Qwen2.5-VL-7B Bai et al. (2025) as the base model for multi-stage granular reinforcement learning training. The training data is primarily derived from Video-R1 Feng et al. (2025). Specifically, we randomly dropout 50% of the CoT annotations in Video-R1-COT-165k for Supervised Fine-Tuning. The Cognition-Aware Refinement dataset comprises 9K

Table 1: Performance of our method on multiple video benchmarks. Vanilla R1 indicates the conventional R1-style GRPO training with our data. CAR and MCT represents Cognition-Aware Refinement and Meta-Cognition Training respectively. For think mode, {N-T, T, A-T} means {Non-Thinking, Thinking, Auto-Thinking}.

| Models (7B) | think mode | Video Perception Benchmarks | | | Video Reasoning Benchmarks | | |
|---|---|---|---|---|---|---|---|
| | | MVBench | TempCompass | VideoMME | VSI-Bench | VideoMMMU | MMVU |
| GPT-4o | N-T | - | - | 71.9 | 34.0 | 61.2 | 75.4 |
| VideoLLaMA2 | N-T | 54.6 | - | 47.9 | - | - | 44.8 |
| LongVA-7B | N-T | - | 56.9 | 52.6 | 29.2 | 23.9 | - |
| LLaVA-OneVision | N-T | 56.7 | - | 58.2 | 32.4 | 33.8 | 49.2 |
| Video-R1 | T | 63.9 | 73.2 | 59.3 | 35.8 | 52.3 | 63.8 |
| Qwen2.5-VL-CoT | T | 59.8 | 72.8 | 57.0 | 32.7 | 50.0 | 59.5 |
| ⊢ +vanilla R1 | T | 64.0 | 73.0 | 60.0 | 38.1 | 52.1 | 65.9 |
| ⊢ +SFT | N-T | 65.2 | 73.7 | **60.9** | 33.1 | 49.1 | 65.2 |
| ⊢ ++CAR | N-T | 65.7 | 74.1 | 59.9 | 37.9 | 50.1 | 65.4 |
| ⊢ +++MCT | **A-T** | **67.2** | **74.4** | 59.7 | **38.6** | **52.3** | **67.3** |
| Thinking Rate | | (76.1%) | (56.2%) | (41.5%) | (100%) | (76.1%) | (19.7%) |

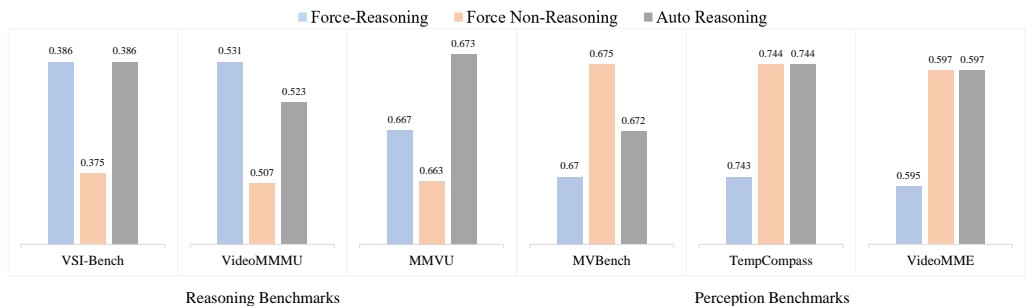

Figure 3: Performance of our final model under different reasoning strategies.

QA pairs, where 8K are selected from Video-R1-260K and 1K are sourced from ScanNet Dai et al. (2017). The Meta-Cognition Training dataset contains 4K samples (also filtered from Video-R1-260K) with a 1:1 ratio of thinking to non-thinking instances. We conduct evaluations across 6 video understanding/reasoning datasets, including VSI-Bench Yang et al. (2025b), VideoMMMU Hu et al. (2025b), VideoMME Fu et al. (2025), MMVU Zhao et al. (2025), TempCompass Liu et al. (2024) and MVBench Li et al. (2024b). Among them, VideoMME, MVBench and TempCompass are more focused on perception, while VSI-Bench, VideoMMMU, and MMVU are knowledge-intensive reasoning benchmarks.

**Training Details.** We train the model in 1 epoch for SFT and Cognition-Aware Refinement and 2 epochs for Meta-Cognition Training. The batchsize per GPU is 1, with $G = 8$ responses are sampled for each query. We utilize 16 H20 GPUs for RL training. For GRPO, we set $\epsilon = 0.2$ and $\beta = 0.04$. The decision advantage for starting tokens is $\gamma = 0.05$, and we set the decision loss probability $p = 0.5$ for Meta-Cognition Training phase. The whole project is built on VeRL framework Sheng et al. (2024) and we will open-source our codes for better reproducibility.

## 4.2 MAIN RESULT

The impact of our multi-stage granular RL scheme on model performance is detailed in Table 1. Overall, based on Qwen2.5VL-7B Bai et al. (2025), we further advance the model's video reasoning capability and for the first time implement a video reasoning model that supports automatic reasoning. Compared with other open-source video-MLLMs Cheng et al. (2024); Zhang et al. (2024); Li et al. (2024a); Feng et al. (2025), our method achieves state-of-the-art (SOTA) results across multi-

Table 2: Performance under different hyperparameter settings, where $\gamma$ scales the meta-cognition advantage and $p$ controls the supervision strength of Eq. 4 and Eq. 5 in Meta-Cognition Training phase.

| $(\gamma, p)$ | Avg T Rate | Video Perception Benchmarks | | | Video Reasoning Benchmarks | | |
|---|---|---|---|---|---|---|---|
| | | MVBench | TempCompass | VideoMME | VSI-Bench | VideoMMMU | MMVU |
| (0.05, 0.5) | 61.6% | 67.2 | 74.4 | 59.7 | 38.6 | 52.3 | 67.3 |
| (0.1, 0.5) | 77.2% | 66.5 | 74.2 | 60.2 | 38.1 | 52.3 | 67.1 |
| (0.2, 0.5) | 81.9% | 67.0 | 74.6 | 59.6 | 38.4 | 52.8 | 65.6 |
| (0.3, 0.5) | 100% | 67.2 | 74.0 | 59.6 | 38.5 | 53.2 | 66.8 |
| (0.1, 0) | 0% | 65.1 | 74.9 | 61.2 | 37.9 | 49.0 | 62.2 |
| (0.1, 0.3) | 90.9% | 65.9 | 74.1 | 59.2 | 39.2 | 50.5 | 65.9 |
| (0.1, 0.7) | 83.4% | 66.9 | 73.2 | 59.7 | 40.0 | 51.4 | 65.1 |
| (0.1, 1) | - | *Training Collapse* | | | *Training Collapse* | | |

ple datasets. Specifically, after the SFT phase with random-CoT-Dropout, the model initially learns two response strategies but adopts the non-thinking mode across all datasets under the rollout=1 test setting. Following the Cognition-Aware Refinement, the model's reasoning and direct response capabilities are both enhanced. However, since no supervision is applied to starting token generation, the model still retains the all-non-thinking strategy. After the completion of Meta-Cognition Training, the model exhibits pronounced automatic reasoning capabilities, displaying drastically divergent thinking rates across different datasets, with a substantially higher thinking rate observed on reasoning benchmarks. Compared to the previous phase, the model after Meta-Cognition Training achieves a substantial performance improvement on all benchmarks. Furthermore, it outperforms the pure thinking model trained via vanilla GRPO Guo et al. (2025) in both response efficiency and answer accuracy. These results fully demonstrate the effectiveness of our method in auto-thinking evocation for video reasoning models.

To further observe the rationality of the model's automatic reasoning decisions, we embedded different starting tokens into prompts to evaluate the auto-thinking model under Force Reasoning and Force Non-Reasoning modes, as shown in Figure 3. On reasoning benchmarks, the accuracy of Non-Reasoning is significantly lower than that of Reasoning mode. Notably, the overall accuracy under automatic reasoning mode shows minimal degradation, with even further performance gains on MMVU. This indicates the model can well distinguish simple questions solvable without reasoning, thus reasonably triggering Non-Reasoning mode while maintaining nearly unchanged overall performance, which accelerates response efficiency. On perception datasets, Non-Reasoning mode outperforms Reasoning mode significantly. This suggests that overthinking for perception-related questions may occasionally lead to errors. Under automatic mode, our model exhibits a notably higher Non-Reasoning trigger rate on perception datasets than on Reasoning datasets. Meanwhile, it achieves better performance than Reasoning mode on perception datasets. These results validate that the model can effectively identify questions that should not reasoning, thereby avoiding the wrong answers caused by overthinking.

## 5 ABLATION STUDIES

In this section, we analyze the impact of hyperparameters on the training process, focusing on the magnitude of Decision advantage $\gamma$ in Eq. 3, the probability parameter $p$ in the gating mechanism of Meta-Cognition Training, and the training order of different stages. Additionally, we present experimental explorations of traditional reward-based approaches for eliciting automatic reasoning, along with the effects of various multi-objective optimization schemes in Meta-Cognition Training. Please refer to supplementary material for more details.

### 5.1 EFFECT OF DIFFERENT $\gamma$ AND $p$.

$\gamma$ controls the magnitude of advantage corresponding to starting tokens. As observed in Table 2, varying $\gamma$ exert minimal impact on response accuracy but exhibit a clear linear relationship with thinking rate: larger value yields higher thinking rates. This is because that the model predom-

Table 3: Performance of our multi-stage granular reinforcement learning under different training orders. MCT indicate Meta-Cognition Training, and {CAR-T, CAR-NT} means GRPO training conditioned on Thinking mode and Non-Thinking mode.

| Train Phase | Avg T Rate | Video Perception Benchmarks | | | Video Reasoning Benchmarks | | |
|---|---|---|---|---|---|---|---|
| | | MVBench | TempCompass | VideoMME | VSI-Bench | VideoMMMU | MMVU |
| ⊢ +SFT | 0% | 65.2 | 73.7 | 60.9 | 33.1 | 49.1 | 65.2 |
| ⊢ ++MCT | 67.1% | 64.7 | 74.4 | 58.6 | 31.8 | 51.2 | 64.8 |
| ⊢ +++CAR-T | 56.3% | 67.0 | 74.6 | 60.1 | 39.1 | 51.7 | 66.3 |
| ⊢ ++++CAR-NT | 48.6% | 67.6 | 74.5 | 60.2 | 38.1 | 52.7 | 66.1 |

inantly generate responses in non-thinking mode in early training, and larger $\gamma$ imposes greater penalties on non-thinking behavior during initial training phases, ultimately elevating the final thinking rate. Based on Table 2 results, $\gamma = 0.05$ achieves better performance. $p$ denotes the probability of selecting the meta-cognition objective during gradient backpropagation. When $p = 0$, Meta-Cognition Training resembles Cognition-Aware Refinement and fails to elicit automatic reasoning. When $p = 1$, Meta-Cognition Training propagates gradients solely for starting tokens, causing the model to gradually forget subsequent token formatting and eventually cause training collapse. Stable automatic reasoning capabilities emerge for $p$=0.3, 0.5, and 0.7, with $p = 0.5$ achieving the optimal overall performance.

### 5.2 PERFORMANCE UNDER DIFFERENT TRAINING ORDERS.

In our main experiments, the training sequence follows SFT → Cognition-Aware Refinement → Meta-Cognition Training. Under this order, the model first acquires preliminary bi-mode response capabilities, then enhances response accuracy for each specific mode, and finally learns to automatically select appropriate reasoning strategies based on the input question. To further investigate how training stage order affects performance, we swap Cognition-Aware Refinement and Meta-Cognition Training, to let the model first learn automatic reasoning before improving mode-specific capabilities. Detailed results are presented in Table 3. Two key observations emerge: first, the model achieves comparable overall performance across different training orders. Second, if automatic reasoning is learned first, subsequent Cognition-Aware Refinement (even without gradient computation for starting tokens) still alters the model's response preference and reduces the overall thinking rate. We attribute this phenomenon to the construction rule of our Meta-Cognition Training data: a question is labeled as "non-thinking-required" if the model generates correct answers in non-thinking mode with high confidence. This enables the model to determine suitable reasoning strategies based on confidence levels. Cognition-Aware Refinement boosts the model's confidence in correct answers, thereby lowering the overall thinking rates.

## 6 CONCLUSION

This paper addresses the critical challenge of evoking automatic reasoning in video reasoning for MLLMs. Specifically, we decouple different functional response tokens and decompose the auto-thinking process into two functionally distinct subtasks: 1) determining the reasoning mode via starting tokens, and 2) generating accurate answers via subsequent tokens. We design a multi-stage granular RL paradigm to respectively enhance the model's capabilities in these two subtasks, which enables stable, controllable, and efficient auto-thinking evocation. In detail, our approach unfolds in three sequential stages: 1) SFT with thinking dropout, which equips the model with bi-mode response capabilities; 2) Cognition-Aware Refinement, which improves mode-specific answer accuracy; 3) Meta-Cognition Training, which explicitly supervises starting token generation using a manually constructed strategy dataset, complemented by a gated multi-objective optimization scheme to mitigate gradient conflicts and prevent output incoherence. Through this fine-grained optimization design, we for the first time achieves controllable and stable auto-thinking activation in video reasoning, which largely avoiding overthinking without compromising accuracy. Extensive experiments across 6 video perception/reasoning benchmarks validate the effectiveness of our method, which achieves new state-of-the-art performance while drastically reducing response overhead.

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

# A   APPENDIX

## A.1   LARGE MODEL USAGE STATEMENT

We only used the large language model to polish our writing, using it after we finished our first draft to refine our expression and make it more coherent. From the discovery of the motivation, the formulation of the idea, the specific code implementation, and the experimental process, we did not use any large language model. Our article does not contain any hidden LLM prompts.

Table 4: GRPO training results (accuracy/thinking rate) on datasets with different hard/easy ratio.

| Hard-Easy | Video Perception Benchmarks | | | Video Reasoning Benchmarks | | |
|---|---|---|---|---|---|---|
| | MVBench | TempCompass | VideoMME | VSI-Bench | VideoMMMU | MMVU |
| 5-5 | 65.4/0% | 74.4/0% | 59.7/0% | 36.4/0% | 4.7/0% | 61.9/0% |
| 3-2 | 65.0/0% | 72.5/0% | 59.6/0% | 34.4/0% | 50.1/0% | 63.3/0% |
| 3-1 | 64.7/10.9% | 68.8/2.3% | 57.2/19.8% | 34.8/2.5% | 46.7/0.5% | 60.9/1.4% |
| 7-1 | 66.8/9.8% | 74.3/0% | 60.9/5.0% | 34.4/0% | 49.0/0% | 64.6/0% |
| 10-0 | 62.8/99.7% | 71.5/100% | 58.4/100% | 33.2/100% | 50.1/97.8% | 63.2/100% |

## A.2   EXPLORATION ON CONVENTIONAL REWARD-BASED AUTO-THINK EVOCATION

Initially in the project, we referenced existing methods Fang et al. (2025); Zhang et al. (2025) and attempted to realize auto-thinking evocation using accuracy-based reward GRPO. Specifically, a reward of 1.1 was assigned for correct answers in non-thinking mode, while a reward of 1.0 was given for correct answers in thinking mode. We conducted experiments using the same SFT model as in the main paper. For training data, questions with pass8 < 0.3 were defined as hard, and those with pass8 > 0.7 as easy. We manually set different hard-easy ratios and selected 8K samples from Video-R1-260K Feng et al. (2025) for training. The model's performance across different training datasets is presented in Table 4. Notably, depending on dataset difficulty, the model either converged to almost all non-thinking or almost all full thinking. It was challenging to achieve reasonable auto-thinking behavior by adjusting the dataset. Furthermore, since the algorithm employed GRPO for training, excessively hard or easy data failed to generate supervision signals. This not only prevented auto-thinking realization but also often led to poor overall performance. These results indicate that existing auto-thinking training schemes based on accuracy rewards cannot be effectively migrated to video reasoning tasks. We also attempted an experimental scheme based on global penalty Lou et al. (2025), which similarly easily fell into fixed response modes and failed to achieve reasonable auto-thinking capabilities. Compared with existing methods, the multi-stage granular RL framework we designed in the main paper outperforms them significantly in terms of performance, stability, and auto-thinking effectiveness.

Table 5: Performance of different implementation for multi-objective optimization in Meta-Cognition Training.

| Method $\gamma = 0.1$ | Avg T Rate | Video Perception Benchmarks | | | Video Reasoning Benchmarks | | |
|---|---|---|---|---|---|---|---|
| | | MVBench | TempCompass | VideoMME | VSI-Bench | VideoMMMU | MMVU |
| Gated | 77.2% | 66.5 | 74.2 | 60.2 | 38.1 | 52.3 | 67.1 |
| Global Average | 58.1% | 65.3 | 73.1 | 59.4 | 38.6 | 50.7 | 65.4 |
| Global Average ($\gamma = 0.2$) | 0% | 65.4 | 73.9 | 59.7 | 37.5 | 49.6 | 65.4 |
| Fusion Objective | 100% | 66.0 | 73.8 | 59.4 | 39.5 | 51.4 | 65.4 |
| Bi-Mode Loss | 83.2% | 66.4 | 74.4 | 59.4 | 37.2 | 51.4 | 64.6 |

A.3  EFFECTS OF DIFFERENT IMPLEMENTATION FOR MULTI-OBJECTIVE OPTIMIZATION IN META-COGNITION TRAINING

For the multi-objective optimization problem in the Meta-Cognition Training stage, we explored various implementation schemes as follows. 1) Global Average advantage: After computing respective advantages for starting tokens and following tokens, we treat them as a whole to calculate the token-level mean average, following the function below.

$$\mathcal{J}_{Meta-Cognition}(\theta) = \mathbb{E}_{x,o_i}\left[\frac{1}{\sum_{i=1}^{G}|o_i|}\sum_{i=1}^{G}\sum_{t=1}^{|o_i|}\mathcal{L}_{i,t}(\theta) - \beta\mathbb{D}_{KL}[\pi_\theta(\cdot|x)||\pi_{ref}(\cdot|x)]\right] \quad (6)$$

2) Fusion objective: Directly summing Eq. 4 and Eq. 5. 3) Bi-Mode Loss (from R-4B Jiang et al. (2025)): Performing group normalization separately for thinking and non-thinking responses within each group. Experimental results are summarized in Table 5. In summary, the global average method also achieved effective auto-thinking capability but exhibited higher parameter sensitivity—converging to full non-reasoning when $\gamma = 0.2$. The fusion objective directly led to all-thinking convergence. Additionally, replacing Eq. 5 in the gating mechanism with Bi-Mode Loss resulted in degraded performance. Overall, the gating mechanism demonstrated the best robustness and overall performance across all schemes.

A.4  CASE STUDY

Our trained auto-thinking model outperforms pure reasoning models across multiple datasets. We present examples where the auto-thinking model avoids over-thinking, as shown in the following figures.

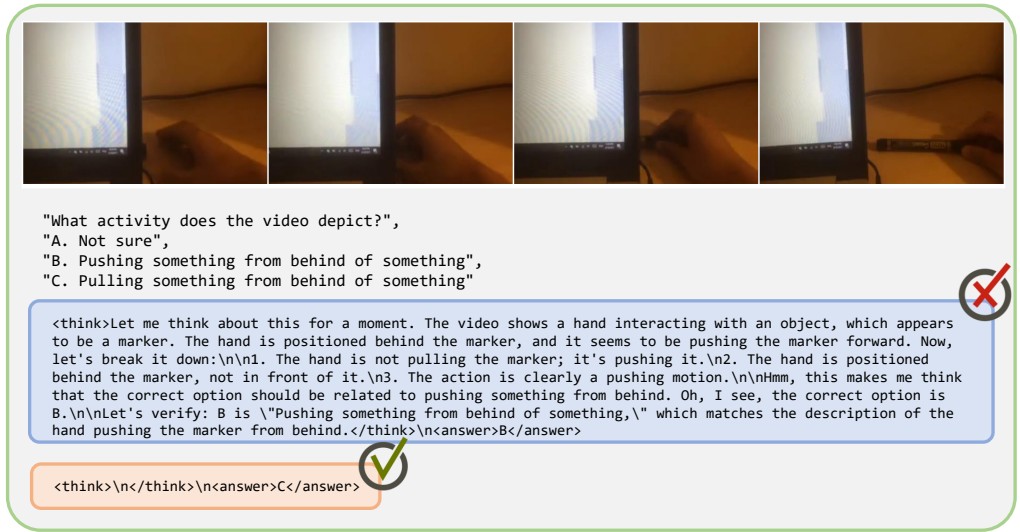

Figure 4: The video shows a hand pulling a pen out from behind a computer. The pure-reasoning model incorrectly infers this process as a "pushing" action, leading to an incorrect answer. In contrast, the auto-thinking model directly generates output in non-reasoning mode and yields the correct answer.

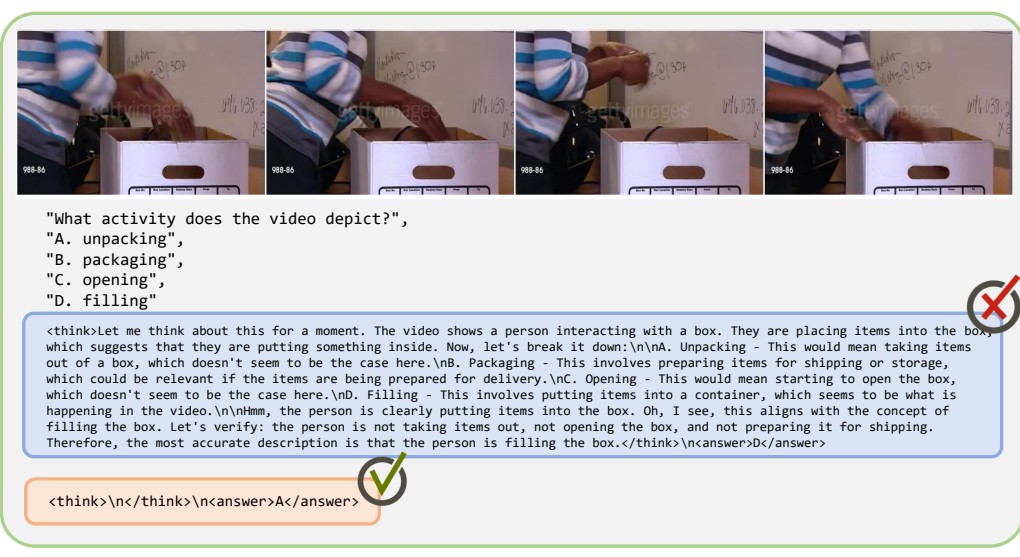

Figure 5: The video shows a person taking items out of a box. The pure-reasoning model incorrectly classifies this process as "putting items into the box", leading to an incorrect answer. By recognizing the question as simple, the auto-thinking model directly generates the correct answer in non-reasoning mode, avoiding mislead by reasoning processes.

