# OpenReview forum: "Auto-Thinking Evocation in Video Reasoning via Multi-Stage Granular Reinforcement Learning: Stable, Controllable"
_ICLR.cc/2026/Conference — ICLR 2026 Conference Withdrawn Submission_

### Official Review · Reviewer_t3i8 · 2025-10-29

**Soundness:** 3
**Presentation:** 3
**Contribution:** 3
**Rating:** 4
**Confidence:** 4

**Summary:**

The paper proposes a multi-stage granular reinforcement learning paradigm to evoke auto-thinking in video-MLLMs while maintaining efficiency. The key idea is to decouple response generation into two subtasks tied to token positions: (1) starting tokens determine the reasoning mode; (2) subsequent tokens determine answer correctness. The training comprises: (a) SFT with “thinking dropout” to initialize bi-mode responses; (b) Cognition-Aware Refinement that refines mode-specific accuracy via GRPO while disabling gradients on the starting tokens to preserve mode preference; (c) Meta-Cognition Training that explicitly supervises starting-token decisions using a strategy dataset labeled by a non-reasoning confidence criterion, plus a gated multi-objective update combining a decision loss (on starting tokens) and a vanilla loss (on subsequent tokens) for stability. Experiments on 6 benchmarks show controllable, dataset-dependent think rates with improved accuracy and better performance-efficiency trade-offs vs. pure-thinking GRPO and prior Video-R1. Ablations analyze hyperparameters, training order, and multi-objective variants, with gating proving most stable. The approach reaches SOTA across several video perception/reasoning tasks using Qwen2.5-VL-7B.

**Strengths:**

- Originality: Clear conceptual decoupling of token roles—starting tokens for mode selection vs. subsequent tokens for correctness—leading to two tailored RL phases (Sec. 3, Fig. 2). Freezing gradients for the initial l tokens during CAR (Sec. 3.2) and the gated multi-objective update in MCT (Sec. 3.3) are simple yet novel designs that address instability and mode collapse seen in unified RL.
- Quality: Comprehensive empirical study across 6 video benchmarks with both perception and reasoning foci (Sec. 4.2, Table 1). Strong ablations: sensitivity to γ and p and their effect on think rate vs. accuracy (Table 2), training order (Table 3), and comparisons of multi-objective optimization schemes showing the robustness of the proposed gating (App. A.3, Table 5). Negative results for conventional reward engineering substantiate the need for the proposed design (App. A.2, Table 4).
- Clarity: The pipeline and token-role decomposition are well illustrated (Figs. 1–2). Mode forcing via starting tokens is clearly explained, as are the decision/vanilla objectives and the gating mechanism (Sec. 3.2–3.3).
- Significance: Demonstrates controllable, dataset-dependent think activation with improved performance-efficiency trade-offs over pure reasoning (Table 1; Fig. 3), addressing a practical issue of over-reasoning in video MLLMs. Provides a generally applicable training recipe that may transfer to other multimodal reasoning tasks (Sec. 1).

**Weaknesses:**

- Limited baseline coverage for auto-thinking in video: While Appendix A.2 analyzes standard accuracy-weighted GRPO variants, comparisons to recent auto-thinking methods adapted to video (e.g., R-4B bi-mode RL [Jiang et al., 2025]) are missing in the main results; only pure-thinking baselines are reported (Table 1). A direct empirical comparison would strengthen claims of stability and controllability.
- Efficiency claims not quantified: The paper argues reduced “response overhead,” but reports think rates rather than concrete token counts, latency, or cost measurements (Secs. 1, 4.2, Fig. 3). Quantifying average output tokens and wall-clock latency under auto vs. forced modes would substantiate efficiency gains.
- Heuristic choices need deeper analysis: The decision dataset labeling hinges on thresholds for non-reasoning confidence (P^gt < 0.5 vs > 0.9; Sec. 3.3). Sensitivity to these thresholds and their impact on mode balance/accuracy is not reported. Similarly, reliance on specific trigger tokens (“Let me think” vs. <think>\n</think>) may be brittle across languages or prompts.
- Scope and generalization: Experiments are primarily multiple-choice; open-ended QA and temporal reasoning with free-form outputs are not evaluated. The method is shown on a single 7B backbone; cross-model scaling and generalization are untested (Sec. 4.1).
- Data construction and filtering: CAR training uses filtered subsets (9k total, with difficulty pruning; Sec. 3.2). More detail on selection biases and their effect on generalization would help. Also, the auxiliary ScanNet 1k spatial data is small; ablation of its contribution is not shown.

**Questions:**

- Please provide concrete efficiency metrics: average generated tokens, decoding latency, and throughput under auto-thinking vs. forced reasoning/non-reasoning across datasets (complementing Fig. 3). Do efficiency gains remain when using larger backbones?
- How sensitive is MCT to the confidence thresholds (0.5/0.9) used to label “thinking required” vs “no required” (Sec. 3.3)? Could you report think rates and accuracy when varying these thresholds or using a learned calibrator?
- Robustness to trigger design: How does the approach behave with different starting phrases, multilingual prompts, or if users prepend their own text before <think>? Any failures where the model outputs the desired starting token but still switches mode mid-generation?
- Can you add a direct baseline with R-4B-style bi-mode annealing and a video reward design, trained on your data and base model, to quantify the stability/controllability advantage (Sec. 2)?
- Open-ended tasks: Does the method transfer to free-form video QA or rationale-required outputs where answer extraction is not multiple-choice? How do you compute rewards for subsequent tokens in those cases?
- Gating mechanism: Could you report gradient conflict metrics (e.g., cosine between objectives) or training variance across seeds to substantiate the stability claim for gated updates vs. fusion/global-average (Sec. 3.3; App. A.3)?
- Data filtering for CAR (Sec. 3.2): How sensitive are results to the 8-sample reward-based filtering and the 9k size? Please add an ablation on CAR data size/hardness and on the contribution of the ScanNet 1k.
- Safety/format adherence: Does the auxiliary objective in MCT (Eq. 5) fully prevent format drift at inference? Any cases where the model emits partial/empty tags leading to evaluation errors?

---

### Official Review · Reviewer_7HJW · 2025-10-30

**Soundness:** 2
**Presentation:** 2
**Contribution:** 3
**Rating:** 4
**Confidence:** 5

**Summary:**

This paper proposes a multi-stage granular reinforcement learning framework to enable stable and controllable auto-thinking in video reasoning with multimodal large language models. The method decouples training into two phases: a Cognition-Aware Refinement stage that optimizes answer accuracy under a fixed reasoning mode, followed by a Meta-Cognition Training stage that leverages a curated strategy dataset to supervise the generation of the initial token and avoid mode collapse. The approach achieves state-of-the-art performance across six video benchmarks while adaptively adjusting thinking rates, thereby significantly improving both efficiency and accuracy without extra inference overhead.

**Strengths:**

1.The paper tackles the critical and highly practical challenge of "over-reasoning" in large reasoning models, avoiding unnecessary reasoning on simple perception questions.

2.The proposed three-stage process (SFT with thinking dropout → Cognition-Aware Refinement → Meta-Cognition Training) is logically structured and easy to follow.

3.To address the limitation of standard RL reward shaping, which often leads to mode collapse, the method introduces explicit supervision via a strategy dataset for the initial token. This enables robust and stable control over thinking rates across multiple benchmarks.

**Weaknesses:**

1.There are some typos and a careful proofreading is suggested. Here are some details:

a）Most of the citation formats in the paper are incorrect. For example, “DeepSeek-R1 Guo et al. (2025)” should be changed to “(DeepSeek-R1 Guo et al. 2025)”

b）In page 1 line 44 of the subsection”INTRODUCTION”:”...significantly improves the responding accuracy for complex problems.": Please change "responding" to "response". "Response accuracy" is the more standard and parallel term.

c）In page 2 line 78 of the subsection”INTRODUCTION”:”...Driving from the above limitations": Please change "Driving from" to "Driven by".

d）In page 2 line 103 of the subsection”INTRODUCTION”:”...Disrupting the model’s inherit automatic thinking strategies": Please change "inherit" to "inherent".

e）In page 6 line 282 of the subsection”META-COGNITION TRAINING”:”Upon the sampeing results": Please change "sampeing" to "sampling".

f）In page 6 line 290 of the subsection”META-COGNITION TRAINING”:”...thereby achieves auto-thinking activation": Please change "achieves" to "achieving".

g）In page 8 line 422 of the subsection”ABLATION STUDIES”:”...ocusing on the magnitude of Decision advantage": "Decision" should not use capital letters.

2.Limited Evaluation of Model Efficiency. While the paper establishes that the method is able to adaptively control the reasoning process, the claim of improved efficiency remains unquantified. The thinking rate serves as an indirect proxy, but direct metrics such as latency, average generation length, or inference throughput are necessary to substantiate the efficiency gains empirically.

3.Missing Evaluation of Training Stability and Preference Controllability. The central claims of improved training stability and preference controllability are currently presented as qualitative observations. For these claims to be convincing, they must be backed with quantitative data. Providing convergence plots and a quantitative analysis of the tuning-behavior correlation is critical for validation.

4.Limited Baseline and Benchmark Coverage in Main Experiments. The current set of baselines and benchmarks is quite limited for the video reasoning domain. Including recent and representative models (e.g., VideoChat2, LLaVA-NEXT-Video) and datasets (e.g., NExT-QA, MSR-VTT) would provide a more balanced and convincing assessment of the proposed method's performance.

[1]Li K, Wang Y, He Y, et al. Mvbench: A comprehensive multi-modal video understanding benchmark[C]//Proceedings of the IEEE/CVF Conference on Computer Vision and Pattern Recognition. 2024: 22195-22206.

[2]Zhang Y, Li B, Liu H, et al. Llava-next: A strong zero-shot video understanding model, April 2024. URL: https://llava-vl.github.io/blog/2024-04-30-llava-next-video/.

[3]Zhang B, Li K, Cheng Z, et al. Videollama 3: Frontier multimodal foundation models for image and video understanding[J]. arXiv preprint arXiv:2501.13106, 2025.

[4]Xiao J, Shang X, Yao A, et al. Next-qa: Next phase of question-answering to explaining temporal actions[C]//Proceedings of the IEEE/CVF conference on computer vision and pattern recognition. 2021: 9777-9786.

[5]Xu J, Mei T, Yao T, et al. Msr-vtt: A large video description dataset for bridging video and language[C]//Proceedings of the IEEE conference on computer vision and pattern recognition. 2016: 5288-5296.

5.Unclear Motivation and Analysis in Ablation Study (Section 5.2). While the ablation study in Section 5.2 (reversing the MCT and CAR phases) is included, its motivation and the hypothesis being tested remain unclear. The observation of comparable performance and a lower thinking rate is noted but not sufficiently interpreted. A clearer rationale for this experimental design and a deeper analysis of the results are needed to draw meaningful conclusions from this ablation.

**Questions:**

1. Could the authors clarify how “reasoning-required” questions are formally defined? It seems that the authors leverage  ”the probability of generating ground-truth answer tokens (Pgt)” as the criterion in the MCT stage. How do the authors justify that the thresholds such as Pgt<0.5 or Pgt>0.9 reliably reflect the need for reasoning, rather than relying on heuristic or intuitive assumptions?

2.It seems that the reasoning mode in the CAR stage is manually assigned for each question. Could the authors clarify whether this assignment is randomized, and whether such manual control affects the model’s ability to autonomously decide reasoning modes in the subsequent MCT stage?

3.To better assess the scientific rigor of the MCT stage, could the authors elaborate on the construction of the strategy dataset? Key details include the rationale behind the confidence thresholds, strategies for maintaining task balance, and the role of manual verification.

4.The manuscript would be strengthened by a deeper investigation into the hyperparameters γ and p. Were any clear trends observed between their values and model performance or the resulting "thinking rate"? This would help clarify whether the chosen configuration is a fragile optimum or represents a robust setting, which is crucial for the method's generalizability.

5.Beyond the quantitative results, the paper would benefit from a discussion of the key insights gained from the multi-stage training dynamics. For instance, are there observable patterns in how reasoning behaviors evolved across stages, or a clear correlation between the initial token selection in MCT and the efficiency/quality of the reasoning trace generated in CAR?

---

### Official Review · Reviewer_RXvx · 2025-11-01

**Soundness:** 3
**Presentation:** 3
**Contribution:** 3
**Rating:** 4
**Confidence:** 2

**Summary:**

This paper proposes a multi-stage granular reinforcement learning framework that decouples thinking/non-thinking mode selection (achieved via starting tokens) from answer generation, inducing MLLMs to produce controllable and stable automatic thinking. Evaluations across video benchmarks demonstrate that compared to the baseline GRPO model, the proposed framework not only achieves higher thinking rates and accuracy across different datasets but also strikes a better balance between performance and efficiency.

**Strengths:**

Motivation and approach: The paper targets the limitation of R1-based reinforcement learning methods, which can over-reason and behave unstably on diverse video reasoning tasks. It proposes a novel decomposition of the pipeline into two subtasks, pattern selection and answer generation, and introduces a well-structured reinforcement learning framework to train them jointly.

Experimental Results: Experiments on multiple video benchmarks show consistent accuracy gains and substantially higher reasoning activation rates, supporting the claim of controllable and efficient reasoning.

Paper Presentation: The paper writing is clear and cohesive. Design choices are well motivated, and the conclusions are supported by ablations.

**Weaknesses:**

A. In experiments section, it remains unclear whether the observed improvements arise primarily from the proposed reinforcement learning design or from the curated “strategy training” dataset introduced for supervision. A controlled comparison or dataset ablation would be helpful to disentangle these factors.

**Questions:**

B. For Cognition-Aware Refinement and Meta-Cognition Training:
B1. Sequence of stages: What is the correct order in the training pipeline? Figure 1 and lines 88–89 suggest MCT should be placed first, while Figure 2 and the Methods section indicate MCT is the final stage. Please make the order explicit and keep it consistent throughout to avoid ambiguity.
B2. Effect of order on performance: Section 5.2 (lines 459–461) claims the swap results are shown in Table 3, but Table 3 appears to report results from progressively adding stages. Please provide explicit results for both orders or clarify how Table 3 reflects the swapped-order experiment.

---

### Official Review · Reviewer_SmGQ · 2025-11-03

**Soundness:** 2
**Presentation:** 2
**Contribution:** 2
**Rating:** 4
**Confidence:** 4

**Summary:**

This paper introduces a multi-stage granular reinforcement learning (RL) paradigm aimed at evoking "auto-thinking" in video reasoning models. The goal is to enable models to adaptively choose between a full-reasoning mode for complex tasks and a non-reasoning mode for simple ones, thereby balancing high performance with high efficiency. The method's core contribution is functionally decoupling the generation process into two subtasks: (1) using the starting token to determine the reasoning mode, and (2) using subsequent tokens to generate the correct answer. The training regimen involves three stages: SFT with thinking dropout, Cognition-Aware Refinement (CAR), and Meta-Cognition Training (MCT). The authors report that this approach achieves new state-of-the-art results on several video benchmarks, demonstrating a superior performance-efficiency trade-off.

**Strengths:**

1. *Novel and Motivated Problem:* The paper addresses a practical and significant limitation in current reasoning models: the "over-reasoning" phenomenon, where models apply computationally expensive reasoning to simple tasks, impairing efficiency. The goal of achieving stable and controllable "auto-thinking" is well-motivated.

2. *Strong Empirical Results:* The method achieves state-of-the-art performance across multiple video reasoning and perception benchmarks (VSI-Bench, VideoMMMU, MMVU, etc.).

3. *Demonstrated Efficiency:* The final model (termed A-T) shows a clear ability to adapt its reasoning, displaying significantly different "thinking rates" (e.g., 100% on VSI-Bench vs. 19.7% on MMVU). This adaptive behavior leads to a reduction in response overhead while improving overall performance compared to a "Thinking-only" model.

4. *Interesting Methodological Design:* The core idea of functionally decoupling the response tokens (starting token for mode selection, subsequent tokens for answer generation) is an innovative approach to tackling the auto-thinking challenge.

**Weaknesses:**

1. *Unsupported Foundational Premise:* The entire multi-stage framework hinges on a critical assumption, stated in lines 158-159, that "the starting token has negligible impact on the accuracy of the final answer but directly determines the model's reasoning strategy, while the following tokens... play a decisive role in ensuring the accuracy...". This functional decoupling is the primary justification for the separate Meta-Cognition Training and Cognition-Aware Refinement stages. However, the paper provides no direct empirical demonstration or analysis to support this claim.

2. *Insufficient Ablation for Algorithmic Modifications:* The proposed method includes several modifications within standard algorithmic components, such as the introduction of 'thinking dropout' in SFT, disabling gradients for starting tokens in CAR, and the gated multi-objective update scheme in the final RL stage. These modifications deviate from vanilla fine-tuning and standard RL, yet their individual efficiency and necessity are not comprehensively proven through ablation studies comparing the modified version against the original/vanilla version of that specific component.

3. *Unjustified Advantage Assignment:* A specific methodological concern is the direct assignment of advantage values ($+1, -1$, etc.) in Equation 3 during Meta-Cognition Training. This is a significant departure from the normalized, group-relative advantages typically used in methods like GRPO. The paper claims this "avoids the advantage vanish issue" but provides no experiments to support this or to compare this ad-hoc assignment against a standard, normalized computation for this subtask.

4. *Lack of Justification for "Reasonable" Thinking Ratios:* The paper presents highly divergent thinking rates across benchmarks (e.g., 100% on VSI-Bench vs. 19.7% on MMVU) and claims these are reasonable. However, this claim is not adequately substantiated. The paper provides no quantitative analysis or detailed qualitative case studies to verify that the model is indeed reasoning only when necessary and not reasoning only when appropriate. The high/low rates are presented as de-facto evidence of success without a deeper validation of their "reasonableness."

5. *Minor Terminological Error:* In line 65, the paper uses the term "self-regressive property". The standard and more widely accepted term in this context is "auto-regressive."

**Questions:**

1. Could the authors provide a direct empirical validation for the core assumption (lines 158-159)? For example, what is the performance (accuracy and thinking rate) of the final model on a reasoning-heavy benchmark (like VSI-Bench) when it is forced to use the non-reasoning mode by manipulating the starting token?

2. Regarding Equation 3, what was the empirical result of using a standard, group-normalized advantage calculation for the $\mathcal{J}_{decision}(\theta)$ objective instead of the direct, unnormalized advantage assignment? Did it empirically lead to the "advantage vanish issue" as hypothesized?

3. How sensitive is the model to the manually constructed strategy dataset used in Meta-Cognition Training? What is the impact of varying the 1:1 ratio of "thinking required" to "no required" samples?

4. Following on the divergent thinking rates, how can the "reasonableness" of these rates be verified? Beyond the aggregate statistics and the case studies in the appendix, could the authors provide a more in-depth qualitative analysis or quantitative breakdown showing examples of questions the model correctly chose to "think" on versus correctly chose not to "think" on? Is there a quantitative way to validate that the 100% rate on VSI-Bench is a truly necessary response to the benchmark's difficulty, and not a mode collapse?

I am willing to adjust the final rating if above weakness and questions are well-addressed.

---

### Note · Authors · 2025-11-12

I have read and agree with the venue's withdrawal policy on behalf of myself and my co-authors.